# Interactive models of communication at the nanoscale using nanoparticles that talk to one another

Antoni Llopis-Lorente[1,2,3], Paula Díez[4], Alfredo Sánchez[4], María D. Marcos[1,2,3], Félix Sancenón[1,2,3], Paloma Martínez-Ruiz[5], Reynaldo Villalonga[4] & Ramón Martínez-Máñez[1,2,3]

'Communication' between abiotic nanoscale chemical systems is an almost-unexplored field with enormous potential. Here we show the design and preparation of a chemical communication system based on enzyme-powered Janus nanoparticles, which mimics an interactive model of communication. Cargo delivery from one nanoparticle is governed by the biunivocal communication with another nanoparticle, which involves two enzymatic processes and the interchange of chemical messengers. The conceptual idea of establishing communication between nanodevices opens the opportunity to develop complex nanoscale systems capable of sharing information and cooperating.

[1] Instituto Interuniversitario de Investigación de Reconocimiento Molecular y Desarrollo Tecnológico (IDM), Universitat Politècnica de València, Universitat de València, Camino de Vera s/n, Valencia 46022, Spain. [2] Departamento de Química, Universitat Politècnica de València. Camino de Vera s/n, Valencia 46022, Spain. [3] CIBER de Bioingeniería, Biomateriales y Nanomedicina (CIBER-BBN), Madrid 28029, Spain. [4] Department of Analytical Chemistry, Faculty of Chemistry, Complutense University of Madrid, Madrid 28040, Spain. [5] Department of Organic Chemistry I, Faculty of Chemistry, Complutense University of Madrid, Madrid 28040, Spain. Correspondence and requests for materials should be addressed to R.V. (email: rvillalonga@quim.ucm.es) or to R.M.-M. (email: rmaez@qim.upv.es).

Nanotechnology has undergone a remarkable growth in recent years, and a large number of nanodevices such as nanomemories[1], nanobatteries[2,3], nanocontainers[4] and nanomotors[5] have already been developed. Nevertheless, communication between human-made nanodevices remains almost unexplored. Experts in telecommunication and computer engineering have already envisioned the interconnection of nanodevices in 'nanonetworks' and their virtually-unlimited applications in different fields[6–8]. At the nanoscale, traditional communication technologies are not applicable given the large size and power requirements of classical transceivers, receivers and other components[9]. In this context, an interesting approach for establishing communication at the nanometric level is to mimic how nature communicates. Chemical or molecular communication[10], based on transmitting and receiving information by means of molecules (chemical messengers), is the communication form used by living organisms. For instance, cells communicate with neighbours by exchanging chemicals[11–13]; neurons propagate, share and process information by using neurotransmitters[14,15]; and insects, bacteria and many mammals use pheromones to communicate with members of their same species. Inspired by biological organisms and cells, scientists have reported, for instance, the use of DNA systems and enzymatic cascades as a tool for information processing and computing[16,17]. Moreover, chemical logic systems based on individual molecules[18,19] and bio-molecular networks[20–22] have also been developed[23–26].

However, despite these interesting advances, communication between nanoparticles has barely been explored. In this scenario, there are many unanswered questions about the construction of nanonetworks that integrate nanoparticles and molecules, such as: Which molecules should be used to encode information? How will these molecules be recognized? How can recognition be converted into propagation of information, and how can this information be reported? Addressing these questions is no trivial matter, and the experimental realization of these systems is still to come.

In communication theory terms, communication can be defined as the process of establishing a connection between two points for information exchange[27]. At one point of the communication process, there is the sender of information, which receives a stimulus to send a message across. The sender converts information into a code and transmits the message through an appropriate medium. The receiver perceives the message and decodes it. Communication models are systematic representations of the process that helps to understand how communication can be done. For example, in the linear model of communication, the message flows in a straight line from the sender to the receiver, and there is no feedback concept. In the more complex interactive model, the sender channels a message to the receiver and the receiver then sends feedback and channels a message to the original sender[28]. In this context, the interactive model is like two linear models piled on top of each other (Fig. 1). Communication is considered effective if it receives the desired result, response or reaction.

To design human-made nanodevices capable of communicating on the nanoscale, here we show a chemical communication process between gated nanoparticles, which mimics the interactive communication model shown in Fig. 1. It employs Janus Au-mesoporous silica gated nanoparticles containing a mesoporous face, loaded with a cargo and capped with stoppers that can be opened in the presence of a specific stimulus[29–35]; and an Au face that is functionalized with different bio-molecules[36,37]. Cargo delivery from one nanoparticle only occurs after biunivocal communication with the second nanoparticle through two enzymatic processes and the exchange of two chemical messengers.

## Results

**Communication system design and operation.** A representation of the communication process we report herein is depicted in Fig. 2. The first nanomachine ($S1_{gal}$) is loaded with $(Ru(bpy)_3)^{2+}$, capped with β-cyclodextrin (β-CD) attached through disulfide bonds to the mesoporous face and functionalized with enzyme β-galactosidase on the Au face. The second nanomachine ($S2_{gox}$) is loaded with N-acetyl-L-cysteine, capped with a pH-responsive β-CD:benzimidazole supramolecular nanovalve on the mesoporous face and functionalized with glucose oxidase (GOx) on the Au face. When $S1_{gal}$ and $S2_{gox}$ are placed together in an aqueous medium, addition of lactose triggers the communication process. First, lactose is hydrolysed by β-galactosidase into galactose and glucose. The produced glucose is transmitted through the aqueous medium towards the Au face of $S2_{gox}$, where it is recognized by glucose oxidase and hydrolysed into gluconic acid (p$K_a$ = 3.6). The generation of gluconic acid induces a local drop in pH that causes the protonation of benzimidazole groups (p$K_a$ = 5.55)[38] on the mesoporous face of $S2_{gox}$ and the dethreading of the supramolecular nanovalve. $S2_{gox}$ uncapping results in the delivery of entrapped N-acetyl-L-cysteine, which diffuses as feedback toward $S1_{gal}$. Finally, N-acetyl-L-cysteine induces the rupture of the disulfide linkages[39] on the mesoporous face of $S1_{gal}$, and $(Ru(bpy)_3)^{2+}$ is released into the medium. Delivery of $(Ru(bpy)_3)^{2+}$ (output) is expected to occur only when the two nanoparticles communicate.

In terms of the interactive model of communication shown in Fig. 1, $S1_{gal}$ nanoparticles (the sender) receive a stimulus (lactose) and transmit information to the receiver ($S2_{gox}$) via a messenger (glucose). At this point the receiver ($S2_{gox}$) perceives the message and operates as a new sender by channelling the message to the original sender ($S1_{gal}$) by using a second chemical messenger (N-acetyl-L-cysteine). Finally the new receiver ($S1_{gal}$) interprets the message, which results in a final desired response (that is, delivery of $(Ru(bpy)_3)^{2+}$).

**Synthesis of the nanodevices.** In order to prepare $S1_{gal}$ and $S2_{gox}$, mesoporous silica (MS) nanoparticles were first obtained by the hydrolysis and condensation of tetraethyl orthosilicate in basic media using n-cetyltrimethylammonium bromide as a template. The surfactant was removed by calcination in air at a high temperature, which yielded the starting mesoporous support. In a second step, gold nanoparticles were prepared according to the Turkevich-Frens method[40,41]. Then, MS nanoparticles were confined at the interface of Pickering emulsion between paraffin and an aqueous face in order to achieve their partial functionalization with (3-mercaptopropyl)trimethoxysilane, on which Au nanoparticles were attached by the formation of S-Au bonds. Paraffin wax was removed by washing the solid with CHCl$_3$, which yielded the initial Janus Au-MS nanoparticles ($S0$). To prepare $S1_{gal}$, the mesoporous face of $S0$ was loaded with $(Ru(bpy)_3)^{2+}$ and was first functionalized with 3-mercaptopropionic acid on the Au face, and later with (3-mercaptopropyl)trimethoxysilane on the mesoporous face. This solid was capped by the attachment of previously modified β-CDs through disulfide linkages, which yielded solid $S1$. Finally, the β-galactosidase enzyme was covalently immobilized by crosslinking the primary amine groups in the enzyme with carboxylic acid moieties on the Au face, which yielded the final nanoparticles $S1_{gal}$. To obtain $S2_{gox}$, firstly a fraction of the $S0$ was functionalized with (3-iodopropyl)trimethoxysilane on the mesoporous face. Then benzimidazole moieties were attached to the iodopropyl moieties through a nucleophilic substitution reaction and the Au-face was functionalized with 3-mercaptopropionic acid. Next, pores were

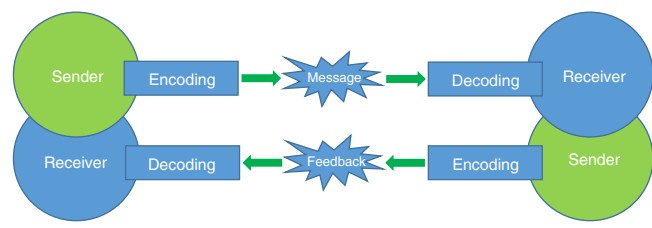

**Figure 1 | Illustration of an interactive model of communication.** The sender receives a stimulus and encodes a message for the receiver. The receiver interprets the message and returns feedback to the first point.

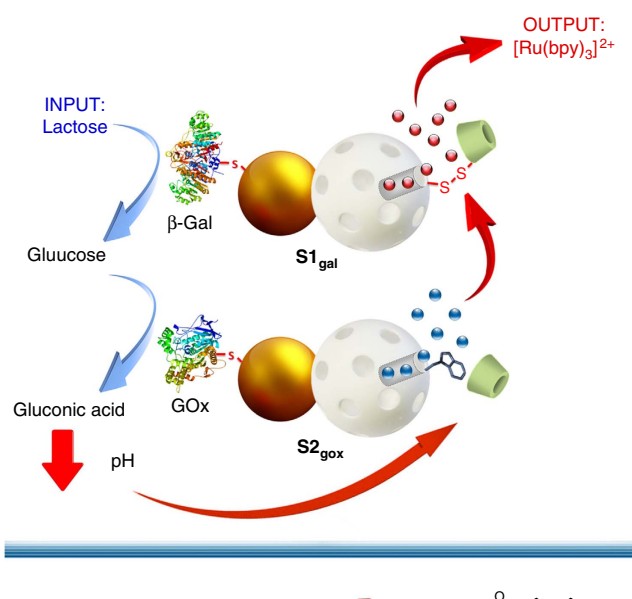

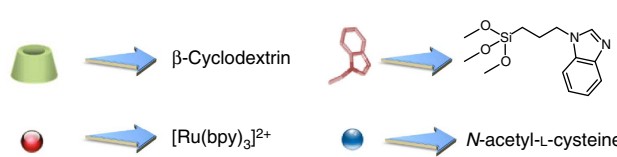

**Figure 2 | Representation of the interactive communication process between two Janus gated nanodevices.** Chemical input (lactose) is hydrolysed by β-galactosidase on **S1gal** to glucose (messenger 1), which is transformed into gluconic acid on **S2gox**. This induces a local drop in pH, which induces the dethreading of the β-CD:benzimidazole supramolecular nanovalve and cargo N-acetyl-L-cysteine release (messenger 2), which is the feedback that finally induces the delivery of the $(Ru(bpy)_3)^{2+}$ reporter from **S1gal**.

loaded with N-acetyl-L-cysteine and capped by the formation of inclusion complexes between benzimidazole groups and β-CDs $(K_{11} = 104\ M^{-1})^{42}$, which resulted in solid **S2**. The glucose oxidase enzyme was then anchored to the Au face using a crosslinking reaction to yield **S2gox**. Furthermore, in order to demonstrate that **S1gal** and **S2gox** are essential for observing communication, **S2blank** was synthesized, which contained the same components as **S2gox**, but lacked the cargo inside the pore voids. In order to assess the capping/uncapping performance of the second nanodevice, we synthesized **S2dye**, which contained the same components as **S2gox** but was loaded with $(Ru(bpy)_3)^{2+}$ which facilitated cargo release monitoring by fluorometric techniques.

**Characterization of the nanodevices.** The different nanoparticles were characterized by standard methods (see Methods and Supplementary Methods for details). The mesoporous

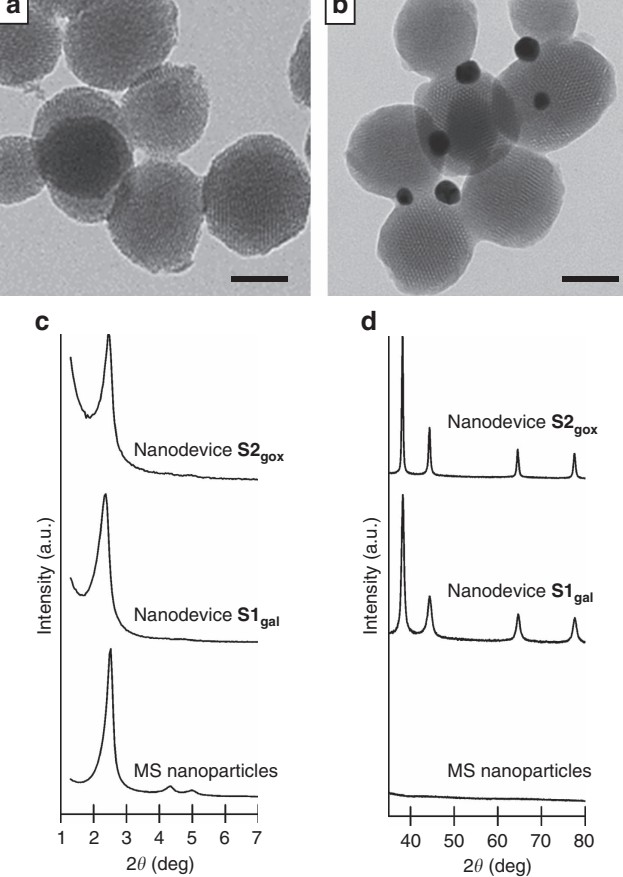

**Figure 3 | TEM images and PXRD of nanoparticles.** TEM of (**a**) calcined MS nanoparticles and (**b**) Janus Au-MS nanoparticles (**S0**) showing the typical porosity of the MCM-41 matrix. Scale bars represent 50 nm. Powder X-ray diffraction pattern of calcined MS nanoparticles, **S1gal** and **S2gox** at low (**c**) and high (**d**) angles. Note the presence of the characteristic gold peaks for Janus colloids around 38°, 44° and 65° and 78°.

morphology of the MS nanoparticles $(81 \pm 18\ nm)$ and the presence of the Au nanoparticles $(19 \pm 4\ nm)$ in Janus colloids **S0** was confirmed by transmission electron microscopy (TEM) analysis (see Fig. 3a,b). The powder X-ray diffraction (PXRD) pattern of the starting MS nanoparticles showed the characteristic mesoporous reflection peak (100) at around 2.4°, which thus confirmed the ordered mesoporous structure (see Fig. 3c). The preservation of this typical peak in the following solids (**S0**, **S1**, **S2**, **S1gal**, **S2gox**) (Supplementary Fig. 1) clearly confirmed that the surface functionalization and cargo loading processes did not damage the mesoporous scaffolding. The diffraction pattern at high angles for all the Janus colloids showed the characteristic cubic gold nanocrystals (111), (200), (220) and (311) peaks[43], which confirmed the Au-MS architecture observed by TEM. The starting gold colloid shows a single absorption band at 520 nm, whereas there is a redshift of the absorbance maximum (535 nm) for **S0** (Supplementary Fig. 2). Regarding the $N_2$ adsorption-desorption isotherms, starting MS and Janus nanoparticles **S0** showed an adsorption step at intermediate $P/P_0$ values (0.1–0.3), which indicates the presence of empty pores in the solid structure (Supplementary Fig. 3). As a result of cargo loading and capping the $N_2$ adsorption-desorption isotherms for the **S1** and **S2** nanoparticles led to a considerable reduction in the $N_2$ volume absorbed, and curves were flat compared to the parent solids (Supplementary Fig. 3). The Brunauer–Emmett–Teller (BET) total specific surface area, pore volumes and pore sizes were

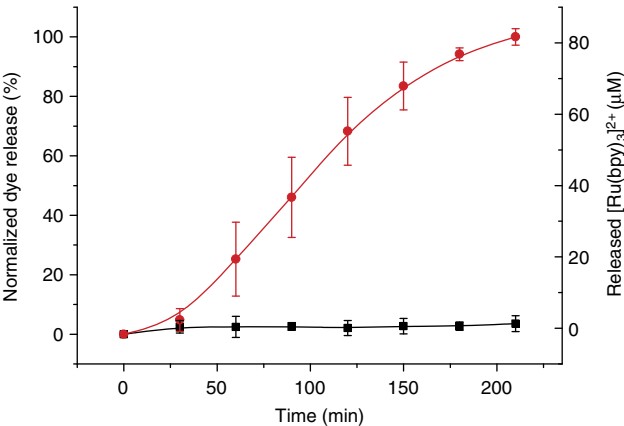

**Figure 4 | Cargo release from the interactive communication between nanoparticles.** $(Ru(bpy)_3)^{2+}$ release in aqueous solution at pH 7.5 that contained $S1_{gal}$ and $S2_{gox}$ in the absence (black curve) and presence (red curve) of lactose (5 mM). Error bars correspond to the s.d. from five independent experiments.

calculated are summarized in Table 1. From the thermogravimetric and elemental analysis studies, the contents of $(Ru(bpy)_3)^{2+}$ and β-CD on **S1** were determined as 119 and 78 mg per gram of solid, respectively (Supplementary Table 1, Supplementary Equations 1–5, and Supplementary Fig. 4). For **S2**, $N$-acetyl-L-cysteine and benzimidazole contents were determined as 31 and 58 mg per gram of solid, respectively. Furthermore, the presence of immobilized enzymes on $S1_{gal}$ and $S2_{gox}$ was confirmed by running specific glucose oxidase and β-galactosidase activity assays on each nanodevice (Supplementary Fig. 5 and Supplementary Equations 6–11).

To investigate the cooperative communication between $S1_{gal}$ and $S2_{dye}$, we confirmed the optimal capping/uncapping behaviour of the molecular gates with the aid of dye-loaded nanoparticles $S1_{gal}$ and $S2_{dye}$ and ultraviolet-visible spectroscopy in the presence or absence of the corresponding messengers. The delivery studies with $S1_{gal}$ revealed that nanoparticles remained capped and displayed no dye release in aqueous solution, whereas the presence of the reducing agent $N$-acetyl-L-cysteine induced the opening of pores and dye release (Supplementary Fig. 6). With $S2_{dye}$, it was confirmed that the $(Ru(bpy)_3)^{2+}$ cargo was released only after adding glucose, whereas the solid remained capped in the absence of the messenger (Supplementary Fig. 7).

**Nanoparticles that talk to one another.** Having characterized the single nanodevices, next we addressed the actual interactive communication process between them. In this complex system, the final release of the reporter $(Ru(bpy)_3)^{2+}$ from nanocarrier $S1_{gal}$ was expected to be related with the information shared between $S1_{gal}$ and $S2_{gox}$ via the exchange of encoding molecules (glucose and $N$-acetyl-L-cysteine). In a typical experiment, $S1_{gal}$ and $S2_{gox}$ were suspended in aqueous solution at pH 7.5 and shaken over time at 25 °C in the presence or absence of 5 mM lactose, which acted as the input signal (see Methods for details). At scheduled times, aliquots were taken, centrifuged to remove nanoparticles, and the absorbance at 453 nm (maximum of the absorption band of $(Ru(bpy)_3)^{2+}$) was measured. Figure 4 shows the time course of the $(Ru(bpy)_3)^{2+}$ delivery from $S1_{gal}$ in the presence and absence of lactose. When there was no input (black curve), no communication between the nanodevices occurred and no output signal was observed. In contrast, when lactose was introduced into the system (red curve), the biunivocal communication in the nanonetwork was triggered, which resulted in the

clear $(Ru(bpy)_3)^{2+}$ reporter release. In the presence of lactose, a total amount of 81 μM of $(Ru(bpy)_3)^{2+}$ was released after 210 min which corresponds to a 39% release efficiency (maximum theoretical release efficiency was calculated by dissolving of $S1_{gal}$ in 20% NaOH solution). The final release of $(Ru(bpy)_3)^{2+}$ was ascribed to an effective interactive model of communication between $S1_{gal}$ and $S2_{gox}$, as indicated in Fig. 2.

For communication to take place, all the individual nanocomponents act together and cooperatively to produce collective behaviour. To demonstrate the crucial role played by the messengers and enzymes in the nanonetwork, additional experiments were carried out with **S1** (lacking β-galactosidase), **S2** (lacking glucose oxidase) and $S2_{blank}$ (lacking the $N$-acetyl-L-cysteine messenger). If in the community $S1_{gal}/S2_{gox}$ the information was shared and the final desired response (that is, delivery of $(Ru(bpy)_3)^{2+}$) was observed, mixtures $S1/S2_{gox}$, $S1_{gal}/S2$ and $S1_{gal}/S2_{blank}$ should not be able to communicate, which would result in no dye release occurring. The delivery experiments with $S1/S2_{gox}$, $S1_{gal}/S2$ and $S1_{gal}/S2_{blank}$ were performed by suspending nanoparticles in an aqueous solution at pH 7.5 in the presence of lactose. In the three uncompleted communities, communication was broken at a certain stage, and the information loop could not close. Therefore, no output signal $((Ru(bpy)_3)^{2+}$ delivery) was observed (see the release profiles shown in Fig. 5). These experiments clearly stress the essential role played by the different system components. Moreover, the operation in the presence of lactose of the community $S1_{gal}/S2_{gox}$ was compared with the system S1/S2/free enzymes, in which enzymes are freely dissolved in the bulk solution. In these experiments, the enzyme-free solids **S1** and **S2** were placed in a solution containing β-galactosidase $(2 U l^{-1})$ and glucose oxidase $(0.8 U ml^{-1})$ (enzymes were dissolved in the bulk solution at an equivalent concentration to that found in the community $S1_{gal}/S2_{gox}$). As can be seen in Fig. 5 (purple curve), in the community S1/S2/free enzymes, the response (that is, delivery of the dye from **S1**) was very low and far from that found for the $S1_{gal}/S2_{gox}$ system. This indicated that in order to have an effective communication, enzymes, in particular glucose oxidase, must be placed in the proximity of the β-CD:benzimidazole complex to be able to generate a local pH drop around the nanoparticle. If the enzymes are in the solution, communication was broken and dye delivery from **S1** was not observed.

A highly desirable characteristic for a communication system is to be selective to a certain input, and it should not respond to other similar inputs that may be in the surroundings. To demonstrate the specificity of the interactive communication between $S1_{gal}$ and $S2_{gox}$ triggered by lactose, the performance of aqueous suspensions of the $S1_{gal}/S2_{gox}$ community in the presence of disaccharides maltose and lactulose was also studied. In these studies, no $(Ru(bpy)_3)^{2+}$ delivery was observed (see Fig. 6). Maltose is not recognized by β-galactosidase on $S1_{gal}$, whereas lactulose is hydrolysed by β-galactosidase into galactose and fructose, but these species are not recognized by glucose oxidase on $S2_{gox}$. In both cases, communication is disrupted and no output signal was found.

Interactive communication is illustrated in Table 2 in a Boolean logic table that indicates how the final output (delivery of $(Ru(bpy)_3)^{2+}$ from $S1_{gal}$) is found only when using the complete community of gated nanoparticles ($S1_{gal}/S2_{gox}$ mixture) in the presence of lactose, but not in its absence or when the sender/receiver is not complete (**S1**, **S2**, $S2_{blank}$, nanoparticles that lack β-galactosidase, glucose oxidase or $N$-acetyl-L-cysteine, respectively).

In conclusion, we have developed an example of communication between nanodevices based on Janus Au-mesoporous silica

**Table 1 | The BET specific surface values, pore volumes and pore sizes calculated from the $N_2$ adsorption-desorption isotherms for the selected materials.**

|  | $S_{BET}$ (m$^2$ g$^{-1}$) | Pore volume* (cm$^3$ g$^{-1}$) | Pore size† (nm) |
|---|---|---|---|
| **MCM-41** | 1093.9 | 0.72 | 2.52 |
| **S0** | 879.1 | 0.62 | 2.33 |
| **S1** | 95.12 | 0.10 | — |
| **S2** | 285.32 | 0.17 | — |

*Total pore volume according to the Barrett–Joyner–Halenda (BJH) model.
†Pore size estimated by using the BJH model applied to the adsorption branch of the isotherm, for $P/P_0 < 0.6$, which can be associated with the surfactant-generated mesopores.

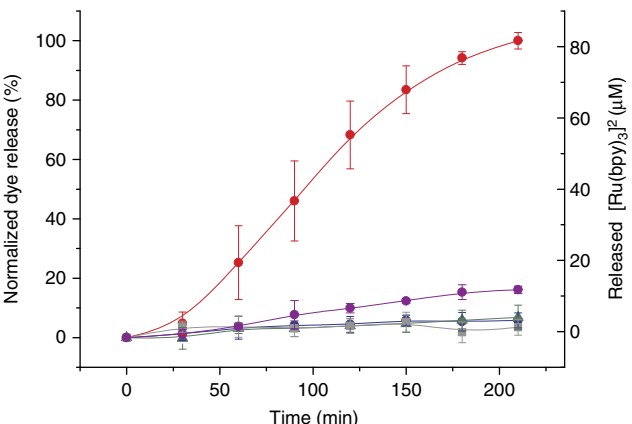

**Figure 5 | Cargo release from uncompleted communities of nanoparticles.** $(Ru(bpy)_3)^{2+}$ release in aqueous solution at pH 7.5 in the presence of lactose (5 mM) that contained communities of nanoparticles **S1 + S2$_{gox}$** (lacking β-galactosidase, blue curve, circles), **S1$_{gal}$ + S2** (lacking glucose oxidase, green curve, triangles), **S1$_{gal}$ + S2$_{blank}$** (lacking the N-acetyl-L-cysteine messenger, grey curve, squares) and **S1 + S2 + free enzymes** (purple curve, circles), (error bars correspond to the s.d. from two independent experiments). Delivery from the full-equipped **S1$_{gal}$ + S2$_{gox}$** system (red curve) is also displayed for comparative purposes. Error bars correspond to the s.d. from five independent experiments. Communication was achieved only when no component was lacking.

nanoparticles which mimics an interactive model of communication, in which a sender nanoparticle receives a stimulus and encodes a message for a receiver nanoparticle that interprets the message and returns feedback to the first nanoparticle. In our communication nanoscale system, delivery from one nanoparticle is governed by an interactive biunivocal communication with another nanoparticle, and involves two enzymatic processes and the use of two chemical messengers. We believe that, in communication terms on the nanoscale, the system we report herein would allow advances to be made in the knowledge of how the recognition of individual molecules (via simple chemical or biochemical reactions) can be used to encode information, and how to convert molecular recognition into information propagation. The idea of establishing communication between nanodevices embraces an enormous potential for the design of more advanced and complex nanoscale systems governed by communication between individual nanocomponents. Inspired by how biological and human communities communicate, the development of such nanodevice communities may open new directions in a number of different areas[44–47].

## Methods

**Synthesis of MCM-41 mesoporous silica nanoparticles.** Approximately 1.00 g (2.74 mmol) of n-cetyltrimethylammonium bromide (CTABr) was dissolved in 480 ml of deionized water. Then, the pH was basified by adding 3.5 ml of a 2 mol l$^{-1}$ NaOH solution and the temperature was increased to 80 °C. Afterward, TEOS (5.00 ml, 22.4 mmol) was added dropwise to this solution. Magnetic stirring

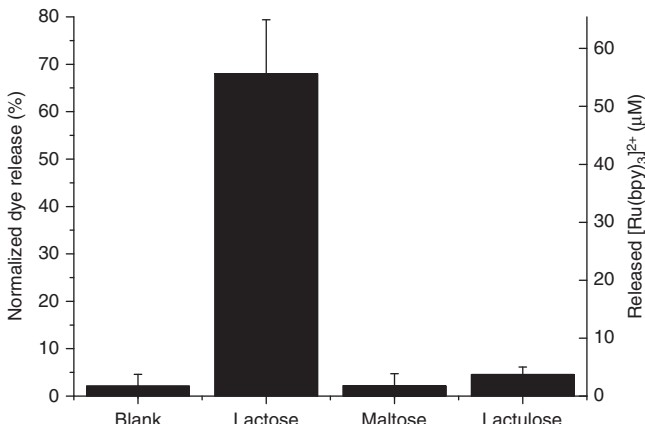

**Figure 6 | Specificity of the interactive communication.** Release of dye in aqueous solution at pH 7.5 for the full-equipped nanonetwork **S1$_{gal}$ + S2$_{gox}$** after 2 h in the presence of three different disaccharides: lactose, maltose and lactulose. Error bars correspond to the s.d. from two independent experiments.

was kept for 2 h to give a white precipitate. Finally, the solid was isolated by centrifugation, washed several times with water and dried at 70 °C overnight (as-synthesized MCM-41). To obtain the final mesoporous nanoparticles (**MCM-41**), the as-synthesized solid was calcined at 550 °C using an oxidant atmosphere for 5 h in order to remove the surfactant.

**Synthesis of gold nanoparticles.** Gold nanoparticles were synthesized based on the Turkevich–Frens method[40,41]. Briefly, 100 ml of a 3 μM HAuCl$_4$·3H$_2$O solution was heated to boiling under stirring and refluxing. Then, 750 μl of a 3.9 μM trisodium citrate solution was added to synthesize 20 nm gold nanoparticles. The initially faint yellow colour turned to blue-black and finally red-wine in 10 min. After this, the colloidal suspension was let to cool at room temperature.

**Synthesis of Janus Au-MS nanoparticles (S0).** Janus nanoparticles were synthesized following a method recently reported by us[36,37]. MCM-41 mesoporous silica nanoparticles (200 mg) were dispersed in 10 ml of aqueous solution (6.7% ethanol) and n-cetyltrimethylammonium bromide (CTABr) was added for a 1 μM final concentration. The mixture was heated at 75 °C, and then 1 g of paraffin wax was added. Once the paraffin was melted, the mixture was vigorously stirred at 25,000 r.p.m. for 10 min using an Ultra-Turrax T-10 homogenizer (IKA, Germany). Afterward, the mixture was further stirred for 1 h at 4,000 r.p.m. and 75 °C using a magnetic stirrer. The resulting Pickering emulsion was then cooled to room temperature, diluted with 10 ml of methanol and reacted with 200 μl of (3-mercaptopropyl) trimethoxysilane. After 3 h under magnetic stirring, the solid was collected by filtration and washed with methanol. For gold attachment, the partially mercapto-functionalized MCM-41 nanoparticles were dispersed in 75 ml of methanol and added over 400 ml of the as-synthesized gold nanoparticles. The mixture was stirred overnight. Then, the solid was isolated by filtration and exhaustively washed with ethanol and with chloroform. The solid was dried and ground. This process finally yielded the Janus Au-MS nanoparticles (**S0**).

**Synthesis of β-CD-S-SO$_2$CH$_3$.** NaSSO$_2$CH$_3$ (70 mg, 0.52 mmol) was added to a solution of mono 6-iodo-6-deoxy-β-cyclodextrin (0.5 g, 4.0 × 10$^{-4}$ mmol) in anhydrous DMF (5 ml) under Argon and warmed to 50 °C while stirring[48]. After 18 h, the solution was cooled and the solvent evaporated. The residue was washed with EtOH (2 × 3 ml) and acetone (2 × 3 ml), yielding 470 mg of a white solid (94%).

**Table 2 | Summary of the response of the communication system when using the full-equipped (S1$_{gal}$ and S2$_{gox}$) or partially-equipped (S1, S2, S2$_{blank}$) nanoparticles and the presence or absence of input (lactose).**

| External trigger[*] (lactose) | Presence of effector 1[*] (β-galactosidase) | Presence of effector 2[*] (GOx) | Presence of messenger 3[*] (NAC) | Response ((Ru(bpy)$_3$)$^{2+}$)[†] |
|---|---|---|---|---|
| 0 | 0 | 0 | 0 | 0 |
| 0 | 1 | 1 | 1 | 0 |
| 1 | 0 | 1 | 1 | 0 |
| 1 | 1 | 0 | 1 | 0 |
| 1 | 1 | 1 | 0 | 0 |
| 1 | 0 | 0 | 0 | 0 |
| 0 | 1 | 0 | 0 | 0 |
| 0 | 0 | 1 | 0 | 0 |
| 0 | 0 | 0 | 1 | 0 |
| 1 | 1 | 0 | 0 | 0 |
| 1 | 0 | 1 | 0 | 0 |
| 0 | 1 | 1 | 0 | 0 |
| 1 | 0 | 0 | 1 | 0 |
| 0 | 1 | 0 | 1 | 0 |
| 0 | 0 | 1 | 1 | 0 |
| 1 | 1 | 1 | 1 | 1 |

*The presence or absence of the trigger (lactose), enzymes (β-galactosidase and glucose oxidase) and N-acetyl-L-cysteine (NAC) in the community of nanoparticles is represented by '1' and '0', respectively.
†Delivery or not of (Ru(bpy)$_3$)$^{2+}$ dye is represented by 0 = no release, 1 = release.

**Synthesis of S1.** Around 50 mg of Janus Au-MS nanoparticles (**S0**) were suspended in a concentrated solution of (Ru(bpy)$_3$)Cl$_2$ (25 mg) in acetonitrile (5 ml), and stirred during 24 h in order to achieve the loading of the pores. Then, the suspension was treated with 50 μl of 3-mercaptopropionic acid for 1 h, filtered, and washed with toluene. Afterwards, this solid was suspended in 6 ml of toluene and reacted with an excess of (3-mercaptopropyl) trimethoxysilane (50 μl) for 24 h. The thiol-functionalized solid was treated with 100 mg of potassium tert-butoxide, isolated by centrifugation and washed with toluene and dimethylformamide (DMF). Finally, the solid was suspended with 50 mg of the as-synthesized β-CD-S-SO$_2$CH$_3$ in 10 ml of DMF under inert atmosphere for 24 h in order to cap the pores. Afterwards, the solid was isolated by centrifugation, washed with DMF and acetronitrile and dried under vacuum. This process resulted in the capped Janus solid **S1**.

**Synthesis of S1$_{gal}$.** Around 20 mg of Janus **S1** were suspended in 10 ml of 50 mM sodium phosphate buffer (pH 7.5) and then 5 mg of β-galactosidase, 5 mg of EDC and 5 mg of N-hydroxysuccinimide (NHS) were added. The mixture was stirred in an ice bath overnight. The coupling reaction is based on the coupling reaction between the amine primary groups of the enzyme and the carboxylic groups on the Au surface. The solid was isolated by centrifugation, washed several times with a cold solution of 50 mM sodium phosphate buffer (pH 7.5) and kept wet in the refrigerator until use. This process yielded the final nanomachine **S1$_{gal}$**.

**Synthesis of S2.** Around 100 mg of **S0** were first suspended in 10 ml of anhydrous acetonitrile under stirring, and then treated with an excess of (3-iodopropyl) trimethoxysilane (100 μl, 0.5 mmol). The suspension was stirred overnight and then the solid was isolated by centrifugation, washed with acetonitrile and dried at 70 °C overnight. To functionalize the surface with benzimidazole moieties, the resulting solid was ground and suspended in 8 ml of a saturated solution of benzimidazole in toluene at 80 °C and 24 ml of triethylamine were then added (toluene and triethylamine in 1:3 v/v ratio). The suspension was stirred and heated at 80 °C for three days. After this, the resulting solid was filtered off, washed with acetonitrile and dried at 70 °C overnight. To protect the gold face, 100 mg of the benzimizadole-functionalized solid was suspended in 8 ml of EtOH and reacted with an excess of 3-mercaptopropionic acid (100 μl). The solid was centrifuged, rinsed with ethanol and with water and let to dry a room temperature for 1day. 100 mg of this solid was suspended in 10 ml aqueous solution of N-acetyl-L-cysteine (0.5 g) at pH 7. After 12 h, 10 ml of 50 mM sodium phosphate buffer containing β-cyclodextrin (1.2 mg) and N-acetyl-L-cysteine (0.250 g) at pH 7.5 were added to the solid suspension and stirred overnight. Finally, the solid was filtered off, washed thoroughly with 50 mM phosphate buffer at pH 7.5 and dried under vacuum for 12 h. This process finally yielded the final solid **S2**.

**Synthesis of S2$_{gox}$.** Around 20 mg of **S2** were suspended in 10 ml of 50 mM sodium phosphate buffer at pH 7.5. Then, 5 mg of EDC, 5 mg of NHS and 5 mg of glucose oxidase were added and the suspension was stirred overnight at 0 °C. The solid was isolated by centrifugation and washed several times with cold 50 mM sodium phosphate buffer (pH 7.5). The resulting **S2$_{gox}$** was kept wet in refrigerator until use.

**Synthesis of S2$_{blank}$.** Solid **S2$_{blank}$** was prepared following the same procedure described for **S2$_{gox}$** but the mesoporous container was not loaded. First, the mesoporous surface on **S0** was modified with benzimidazole moieties and the gold surface was protected with 3-mercaptopropionic acid as described above. 10 mg of this solid was suspended in 5 ml of 50 mM sodium phosphate buffer at pH 7.5 containing β-cyclodextrin (1.2 mg ml$^{-1}$) and stirred overnight. Then, the solid was filtered off, washed with 50 mM phosphate buffer (pH 7.5) and dried under vacuum for 12 h. To functionalize this solid with the enzyme, we followed the same procedure, as described above. The solid was suspended in 5 ml of 50 mM sodium phosphate buffer (pH 7.5) and 2.5 mg of EDC, 2.5 mg of NHS and 2.5 mg of glucose oxidase were added. The mixture was stirred overnight at 0 °C. Finally, the nanoparticles were isolated by centrifugation and washed several times with cold 50 mM sodium phosphate buffer (pH 7.5). The resulting **S2$_{blank}$** was kept wet in the refrigerator until use.

**Synthesis of S2$_{dye}$.** Solid **S2$_{dye}$** was prepared following the same procedure described for **S2$_{gox}$** but the mesoporous container was loaded with (Ru(bpy)$_3$)$^{2+}$. First, the mesoporous surface on **S0** was modified with benzimidazole moieties and the gold surface was protected with 3-mercaptopropionic acid, as described above. 10 mg of this solid was suspended in 5 ml of aqueous solution of (Ru(bpy)$_3$)$^{2+}$ (5 mg). After 12 h, 10 ml of 50 mM sodium phosphate buffer at pH 7.5 containing β-cyclodextrin (1.2 mg ml$^{-1}$) were added to the solid suspension and stirred overnight. Then, the solid was filtered off, washed thoroughly with 50 mM phosphate buffer at pH 7.5 and dried under vacuum for 12 h. Finally, we follow the same procedure, as described above in order to attach the enzyme glucose oxidase to the Au face. This process finally yields the solid **S2$_{dye}$** that was kept wet in refrigerator until use.

**Chemical communication studies.** To demonstrate the performance of the nanonetwork, the refrigerated supensions of nanoparticles were aliquoted, washed separately with aqueous solution (20 mM Na$_2$SO$_4$) at pH 7.5 and placed together in the same recipient. In a typical experiment, **S1$_{gal}$** and **S2$_{gox}$** (4 mg ml$^{-1}$ and 1 mg ml$^{-1}$, respectively) were suspended together and shaken overtime at 25 °C in the presence or absence of 5 mM lactose, which acts as the input signal. Aliquots were taken at scheduled times, centrifuged to remove the nanoparticles and the absorbance at 453 nm corresponding to the (Ru(bpy)$_3$)Cl$_2$ released was measured. In order to demonstrate the crucial role played by the components of the system, the same procedure was followed with suspensions of **S1/S2$_{gox}$**, **S1$_{gal}$/S2**, and **S1$_{gal}$/S2$_{blank}$**. For the release experiments with the community **S1/S2**/free enzymes, **S1** and **S2** (4 mg ml$^{-1}$ and 1 mg ml$^{-1}$ respectively) were placed in a recipient with Gal (2 U l$^{-1}$) and GOx (0.8 U ml$^{-1}$). For specificity experiments, the procedure was the same but maltose (5 mM) and lactulose (5 mM), instead of lactose, were added as the inputs to suspensions of **S1$_{gal}$/S2$_{gox}$**.

**Data availability.** The authors declare that data supporting the findings of this study are available within the paper and its Supplementary information files. All other relevant data are available from the corresponding authors on request.

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

## Acknowledgements

A.L.-L. is grateful to 'La Caixa' Banking Foundation for his PhD fellowship. We wish to thank the Spanish Government (MINECO Projects MAT2015-64139-C4-1, CTQ2014-58989-P and CTQ2015-71936-REDT and AGL2015-70235-C2-2-R) and the Generalitat Valenciana (Project PROMETEOII/2014/047) for support. The Comunidad de Madrid (S2013/MIT-3029, Programme NANOAVANSENS) is also gratefully acknowledged.

## Authors contributions

A.L.-L. and P.D. performed the experiments. A.L.-L., P.D., A.S., R.V. and R.M.-M. designed and conceived the experiments. A.L.-L. and R.M.-M. wrote the manuscript. F.S. revised and helped in the elaboration of the manuscript. M.D.M. and P.M.-R. helped in useful discussions and technical issues. All authors discussed the results and commented on the manuscript.

## Additional information

**Competing interests:** The authors declare no competing financial interests.

