## [Peer Review File · Nature Communications]

Reviewers' comments:

Reviewer #1 (Remarks to the Author):

This is a really interesting study that describes the preparation of two types of gold-silica nanoparticle (GSN) systems. Each GSN is conjugated to a different enzyme and loaded with different guests. The addition of a stimulus (lactose) triggers a series of coupled chemical transformations that results in the release of Ru(bpy)₃. The process was followed with UV-Vis spectroscopy. In general, the authors demonstrate a successful control of chemical information in a chemical system at the nano size level, which is akin to biological systems. I suggest for this study to be published in Nature Communications as it has been well done with the results of interest to the broad readership of the journal.

Here are my comments:

(1) What is the binding affinity of the cyclodextrin toward benzimidazole (K_d)?

(2) As the experiments were completed in water, how much does the pH change upon the formation of gluconic acid? The related acid could, perhaps, directly protonate the benzimidazole?

(3) Can you please explain the disulfide exchange more closely? I believe that this reaction is fast at basic conditions but quite slow under acidic?

Reviewer #2 (Remarks to the Author):

This manuscript describes an artificial system displaying chemical communication between enzyme and artificial components that are tethered to two populations of nanoparticle. The main claim is that this system represents an "interactive" mode of communication "on the nanoscale" where a message is transmitted from sender to receiver; and in response a second message from receiver back to sender, giving an identifiable output.

The system combines four stimuli-responsive chemical components: two enzyme catalysts, a pH-dependent host-guest complex, and a dynamic covalent disulfide bond. Each of these in isolation is a well-characterized functional molecular system. The nanoscale aspect arises as a result of tethering these molecular components to nanoscale scaffolds created from mesoporous silica nanoparticles.

Two populations of nanoparticle are thus prepared (S1_Gal and S2_GOx). Each of these has one of the two enzymes attached, while one of the abiotic stimuli-responsive components controls guest capture and release from the silica mesopores.

The successful operation of these four components to send a signal from S1_Gal to S2_GOx in the form of an enzyme product/substrate (glucose); and back again from S2_GOx to S1_Gal in the form of a cargo released from S2 that triggers release of a different cargo from S1 – as described in Scheme 2 – is well-supported by the experimental results on the full system, together with a comprehensive set of control experiments. There are two clarifications which the authors need to address as regards their experimental results, detailed under "other points" below.

The experimental evidence supports the operation of an artificial chemical logic device as represented in Table 1 – the 'triggers', 'effectors' and 'messengers' described must all be present in the system to have the 'output' of dye release. However, chemical logic systems have been investigated for many years, based on individual molecules [eg 10.1038/35018259; 10.1002/chem.200305054],

(bio)molecular networks [10.1038/429351a; 10.1126/science.1132493; 10.1126/science.1214081] and nanoparticle systems [eg 10.1021/ja9042752; 10.1021/ja401000m; 10.1039/C4CC08541H; 10.1002/anie.201008198]. This needs to be reflected more accurately in the citations.

The realization of artificial systems that exploit chemical communication is certainly a challenge of current high interest, and going about this using nanoparticles presents several advantages, exciting possibilities, and parallels with "nanoscale" biological systems such as cells. The authors themselves have already published a very nice example of communication between mesoporous nanoparticles in a sequential fashion (reference 51) akin to the S2 → S1 process reported in the new manuscript. The new system represents an advance in terms of complexity over that previous report (including enzyme-based signalling in addition to the abiotic triggers/effectors), but the important question is whether it goes conceptually beyond that of the prior art and really does represent "interactive...communication at the nanoscale". As it stands, I am not convinced.

It is clear that communication on the nanoscale is achieved by releasing cysteine cargo from nanocarrier S2_Gal to trigger release of a different cargo (Ru dye) from nanocarrier S1_GOx. This is clearly a function of communication between nanoparticles. But to claim the "interactive" or feedback model of communication on the nanoscale, the question must be addressed as to what effect is there of tethering the enzymes Gal and GOx on S1 and S2, respectively? The authors have nicely demonstrated that there is no significant effect on either enzyme's operation by confinement to the nanoscale surface. It would therefore appear that these enzymes could equally be added to the system as freely dissolved components in bulk solution, and the same behavior would occur: Gal would hydrolyze lactose to glucose; GOx would oxidize glucose to gluconic acid, and the reduction in pH would trigger the communication between the mesoporous carriers S1 and S2.

If it can be shown that **tethering of GOx to S2 is crucial for the operation of the signalling system** then the authors claims would be more convincing. The key question therefore is: **must the GOx be localized in the proximity of the benzimidazole hosts on the nanoscale** in order to generate only a local drop in pH and triggering the cascade? This would appear to be entirely reasonable but the **key control experiment where (enzyme-free) S1 and S2 are combined with Gal and GOx in freely dissolved bulk solution** has not been reported.

Other points

1. The authors indulge in a lengthy introduction, ranging from the status of nanotechnology as a field in general, through discussion of non-chemical communications technology, and (several examples of) chemical communication in biology. They would do their work better justice by focusing in on the unique aspects demonstrated by the current study.

2. Lines 158-161 describe the results of "thermogravimetric and elemental analysis studies" that allow the calculation of Ru, CD, cysteine and benzimidazole loadings on the respective NP systems. These results, and the corresponding calculations should be included in the SI. While the determination of eg Ru content would appear straightforward by elemental analysis, it is not so clear how the loadings of organic species were determined, given that each of S1 and S2 contains several C/H/N-containing moieties.

3. What proportion of the total dye loaded is released during the 200 min 'signalling' experiments? Having determined the loadings discussed in point 2, it should be possible to calculate the maximum theoretical quantity of Ru that might be released by S1_Gal. This information is not given; "dye released" in Figures 2, 3, and 4 appears to be normalized to "100 %" at the final timepoint recorded. Does this also correspond to 100% of what was loaded? Or some smaller concentration?

4. The caption to Figure 2 identifies the error bars as "from 5 independent experiments", but not what statistic is calculated from these repeats: one standard deviation; two standard deviations; some other measure of confidence/error?

Reviewer 1 remarks to the author

This is a really interesting study that describes the preparation of two types of gold-silica nanoparticle (GSN) systems. Each GSN is conjugated to a different enzyme and loaded with different guests. The addition of a stimulus (lactose) triggers a series of coupled chemical transformations that results in the release of Ru(bpy)₃. The process was followed with UV-Vis spectroscopy. In general, the authors demonstrate a successful control of chemical information in a chemical system at the nano size level, which is akin to biological systems. I suggest for this study to be published in Nature Communications as it has been well done with the results of interest to the broad readership of the journal.

Here are my comments:

1. What is the binding affinity of the cyclodextrin toward benzimidazole (K_d)?
2. As the experiments were completed in water, how much does the pH change upon the formation of gluconic acid? The related acid could, perhaps, directly protonate the benzimidazole?
3. Can you please explain the disulfide exchange more closely? I believe that this reaction is fast at basic conditions but quite slow under acidic?

Author's response to reviewer 1

We thank reviewer 1 for his/her comments and suggestions. Please find below a point-by-point response to the referee's comments.

1. "What is the binding affinity of the cyclodextrin toward benzimidazole?"

The inclusion complex formation between benzimidazole and β -cyclodextrin has been reported to have a complex formation constant of $104 \pm 8 \text{ M}^{-1}$. We have included this information in the manuscript (see page 5 and reference 42). (Yousef, F. O., Zughul, M. B. & Badwan, A. A. The modes of complexation of benzimidazole with aqueous β -cyclodextrin explored by phase solubility, potentiometric titration, ¹H-NMR and molecular modeling studies. J. Incl. Phenom. Macrocycl. Chem. 57, 519–523 (2007)).

2. "As the experiments were completed in water, how much does the pH change upon the formation of gluconic acid? The related acid could, perhaps, directly protonate the benzimidazole?"

We determined the pH of the aqueous suspensions of the **S1_{gal}/S2_{gox}** before and after addition of lactose and it was observed that the pH of the bulk solution did not change appreciably. We tentatively attributed that to the fact that the generated protons produced in **S2_{gox}** are trapped by the benzimidazole moieties (that protonate). Moreover, it can also be taken into account that there are some other protonable groups including silanolates on the silica surface and amine or acidic residues in the enzymes that can also be partially protonated. We believe that delivery in the **S2_{gox}** nanoparticles is due to a local change of the pH (*vide infra*).

3. “Can you please explain the disulfide exchange more closely? I believe that this reaction is fast at basic conditions but quite slow under acidic?”

As pointed by the reviewer, the disulfide exchange has been reported to be slower under acidic conditions. The classical thiol–disulfide interchange reaction is a nucleophilic substitution of a thiol in disulfides with another thiol. From a kinetic point of view, there are several factors that influence the reaction such as the pK_a and nucleophilicity of the attacking thiol, the stability of the leaving group, the electrophilicity of the central disulfide sulfur or the presence of other ionizable groups in the molecule. For example, in our particular case the carboxylic group in the N-acetyl-cysteine molecule is expected to enhance the nucleophilicity of the thiol and the thiolated- β -cyclodextrin is expected to be a good leaving group. (see Nagy, P. Kinetics and mechanisms of thiol–disulfide exchange covering direct substitution and thiol oxidation-mediated pathways. *Antioxid. Redox Signal.* 18,1623-1641 (2013)).

From another point of view, the reduction of disulfide bonds in proteins at pH 7.5 using thiol-containing reducing agents such as 2-mercaptoethanol or DTT (dithiothreitol) is well-extended and known to work finely (see for example Jocelyn, P. C. Chemical reduction of disulfides. *Methods Enzymol.* 143, 246-256 (1987)). Moreover, in the “communication” experiments the pH is not acid and therefore the reaction is expected to be relatively fast.

Reviewer 2 remarks to the author

This manuscript describes an artificial system displaying chemical communication between enzyme and artificial components that are tethered to two populations of nanoparticle. The main claim is that this system represents an “interactive” mode of communication “on the nanoscale” where a message is transmitted from sender to receiver; and in response a second message from receiver back to sender, giving an identifiable output.

The system combines four stimuli-responsive chemical components: two enzyme catalysts, a pH-dependent host–guest complex, and a dynamic covalent disulfide bond. Each of these in isolation is a well-characterized functional molecular system. The nanoscale aspect arises as a result of tethering these molecular components to nanoscale scaffolds created from mesoporous silica nanoparticles.

Two populations of nanoparticle are thus prepared ($S1_{Gal}$ and $S2_{GOx}$). Each of these has one of the two enzymes attached, while one of the abiotic stimuli-responsive components controls guest capture and release from the silica mesopores.

The successful operation of these four components to send a signal from $S1_{Gal}$ to $S2_{GOx}$ in the form of an enzyme product/substrate (glucose); and back again from $S2_{GOx}$ to $S1_{Gal}$ in the form of a cargo released from $S2$ that triggers release of a different cargo from $S1$ – as described in Scheme 2 – is well-supported by the experimental results on the full system, together with a comprehensive set of control experiments. There are two clarifications which the authors need to address as regards their experimental results, detailed under “other points” below.

The experimental evidence supports the operation of an artificial chemical logic device as represented in Table 1 – the ‘triggers’, ‘effectors’ and ‘messengers’ described must all be present in the system to have the ‘output’ of dye release. However, chemical logic systems have been investigated for many years, based on individual molecules [eg 10.1038/35018259; 10.1002/chem.200305054], (bio)molecular networks [10.1038/429351a; 10.1126/science.1132493; 10.1126/science.1214081] and nanoparticle systems [eg 10.1021/ja9042752; 10.1021/ja401000m; 10.1039/C4CC08541H; 10.1002/anie.201008198]. This needs to be reflected more accurately in the citations.

The realization of artificial systems that exploit chemical communication is certainly a challenge of current high interest, and going about this using nanoparticles presents several advantages, exciting possibilities, and parallels with “nanoscale” biological systems such as cells. The authors themselves have already published a very nice example of communication between mesoporous nanoparticles in a sequential fashion (reference 51) akin to the S2 → S1 process reported in the new manuscript. The new system represents an advance in terms of complexity over that previous report (including enzyme-based signaling in addition to the abiotic triggers/effectors), but the important question is whether it goes conceptually beyond that of the prior art and really does represent “interactive...communication at the nanoscale”. As it stands, I am not convinced.

It is clear that communication on the nanoscale is achieved by releasing cysteine cargo from nanocarrier S2_{Gal} to trigger release of a different cargo (Ru dye) from nanocarrier S1_{GOx}. This is clearly a function of communication between nanoparticles. But to claim the “interactive” or feedback model of communication on the nanoscale, the question must be addressed as to what effect is there of tethering the enzymes Gal and GOx on S1 and S2, respectively? The authors have nicely demonstrated that there is no significant effect on either enzyme’s operation by confinement to the nanoscale surface. It would therefore appear that these enzymes could equally be added to the system as freely dissolved components in bulk solution, and the same behavior would occur: Gal would hydrolyze lactose to glucose; GOx would oxidize glucose to gluconic acid, and the reduction in pH would trigger the communication between the mesoporous carriers S1 and S2.

If it can be shown that tethering of GOx to S2 is crucial for the operation of the signaling system then the authors claims would be more convincing. The key question therefore is: must the GOx be localized in the proximity of the benzimidazole hosts on the nanoscale in order to generate only a local drop in pH and triggering the cascade? This would appear to be entirely reasonable but the key control experiment where (enzyme-free) S1 and S2 are combined with Gal and GOx in freely dissolved bulk solution has not been reported.

Other points

1. The authors indulge in a lengthy introduction, ranging from the status of nanotechnology as a field in general, through discussion of non-chemical communications technology, and (several examples of) chemical communication in biology. They would do their work better justice by focusing in on the unique aspects demonstrated by the current study.

2. Lines 158-161 describe the results of “thermogravimetric and elemental analysis studies” that allow the calculation of Ru, CD, cysteine and benzimidazole loadings on the respective NP systems. These results, and the corresponding calculations should be included in the SI. While the determination of eg Ru content would appear straightforward by elemental analysis, it is not so clear how the loadings of organic species were determined, given that each of S1 and S2 contains several C/H/N-containing moieties.

3. What proportion of the total dye loaded is released during the 200 min ‘signaling’ experiments? Having determined the loadings discussed in point 2, it should be possible to calculate the maximum theoretical quantity of Ru that might be released by S1_{Gal}. This information is not given; “dye released” in Figures 2, 3, and 4 appears to be normalized to “100 %” at the final time-point recorded. Does this also correspond to 100% of what was loaded? Or some smaller concentration?

4. The caption to Figure 2 identifies the error bars as “from 5 independent experiments”, but not what statistic is calculated from these repeats: one standard deviation; two standard deviations; some other measure of confidence/error?

Author’s response to reviewer 2

We thank reviewer 2 for his/her comments and suggestions. Please find below a point-by-point response to the referee’s comments.

1. *“Must the GOx be localized in the proximity of the benzimidazole hosts on the nanoscale in order to generate only a local drop in pH and triggering the cascade? This would appear to be entirely reasonable but the key control experiment where (enzyme-free) S1 and S2 are combined with Gal and GOx in freely dissolved bulk solution has not been reported.*

As suggested, we have performed this key control experiment indicated by the reviewer where (enzyme-free) **S1** and **S2** are combined with Gal and GOx which were freely dissolved in the bulk mixture. In particular, we have compared the operation of the community **S1_{gal}/S2_{gox}** with the systems **S1/S2/free** enzymes (in which enzymes are dissolved in the solution). In these experiments, the enzyme-free solids **S1** and **S2** were placed in a solution containing β -galactosidase ($2 \text{ U}\cdot\text{L}^{-1}$) and glucose oxidase ($0.8 \text{ U}\cdot\text{mL}^{-1}$) (enzymes were dissolved in the solution in an equivalent concentration to that found in the community **S1_{gal}/S2_{gox}**). In the system **S1/S2/free** enzymes the response (i.e delivery of the dye from **S1**) was not observed, which indicated that in order to have an effective communication glucose oxidase must be attached in the proximity of the β -CD:benzimidazol complex to be able to generate a local pH drop around the nanoparticle. If the enzymes are in the solution, communication was broken and dye delivery from **S1** was not observed. We have included this new experiment in the corrected version of the manuscript and have provided the corresponding release curve (see Figure 3).

2. *“The authors indulge in a lengthy introduction, ranging from the status of nanotechnology as a field in general, through discussion of non-chemical communications technology, and (several examples of) chemical communication in*

biology. They would do their work better justice by focusing in on the unique aspects demonstrated by the current study.”

Taking into account the reviewer suggestion, we have shortened the introduction in this new version of the journal. The introduction is now a third shorter. We have also included the references suggested by the reviewer about chemical logic systems (see new references 18-26)

3. *“Lines 158-161 describe the results of “thermogravimetric and elemental analysis studies” that allow the calculation of Ru, CD, cysteine and benzimidazole loadings on the respective NP systems. These results, and the corresponding calculations should be included in the SI. While the determination of eg Ru content would appear straightforward by elemental analysis, it is not so clear how the loadings of organic species were determined, given that each of S1 and S2 contains several C/H/N-containing moieties.”*

As suggested by the reviewer, results of elemental and thermogravimetric analysis have been included in the Supporting Information as well as the calculations performed to estimate the contents of each component (see SI page 9).

4. *“What proportion of the total dye loaded is released during the 200 min ‘signalling’ experiments? Having determined the loadings discussed in point 2, it should be possible to calculate the maximum theoretical quantity of Ru that might be released by S1_Gal. This information is not given; “dye released” in Figures 2, 3, and 4 appears to be normalized to “100 %” at the final timepoint recorded. Does this also correspond to 100% of what was loaded? Or some smaller concentration?”*

Taking into account that the communication experiments had been followed by UV-vis spectrophotometry, the amount of $[\text{Ru}(\text{bpy})_3]^{2+}$ released has been calculated by applying the Beer-Lambert Law. In the manuscript, we have included an additional axis in the release plots indicating the amount of $[\text{Ru}(\text{bpy})_3]^{2+}$ released. The normalized “100 %” maximum release for the community **S1_{gal}/S2_{gox}** after 210 min in the presence of lactose corresponds to 81 μM $[\text{Ru}(\text{bpy})_3]^{2+}$, which represents a 39% of delivery efficiency. This amount is similar to other reported gated materials loaded with $[\text{Ru}(\text{bpy})_3]^{2+}$. We have included all this information in the corrected version of the manuscript.

5. *“The caption to Figure 2 identifies the error bars as “from 5 independent experiments”, but not what statistic is calculated from these repeats: one standard deviation; two standard deviations; some other measure of confidence/error?”*

As indicated by the reviewer, we have included in the manuscript that the error bars correspond to one standard deviation.

REVIEWERS' COMMENTS:

Reviewer #2 (Remarks to the Author):

The authors have comprehensively addressed all of the points raised by both reviewers. I congratulate them on an innovative system and very nice manuscript, which should now be accepted for publication in Nature Communications.